# Is the Rationale of Anatomical Liver Resection for Hepatocellular Carcinoma Universally Adoptable? A Hypothesis-Driven Review

**DOI:** 10.3390/medicina57020131

**Published:** 2021-02-02

**Authors:** Young-Jen Lin, Cheng-Maw Ho

**Affiliations:** Department of Surgery, National Taiwan University Hospital and College of Medicine, 7 Chung-Shan South Road, Taipei 100, Taiwan; young332@gmail.com

**Keywords:** anatomical resection, hepatocellular carcinoma, neoangiogenesis, tumor microenvironment, resection margin, circulating tumor cells, protumorigenic niche

## Abstract

Surgical resection is the first-line curative treatment modality for resectable hepatocellular carcinoma (HCC). Anatomical resection (AR), described as systematic removal of a liver segment confined by tumor-bearing portal tributaries, may improve survival by reducing the risk of tumor recurrence compared with non-AR. In this article, we propose the rationale for AR and its universal adoption by providing supporting evidence from the advanced understanding of a tumor microenvironment and accumulating clinical experiences of locoregional tumor ablation therapeutics. AR may be advantageous because it completely removes the en-bloc by interrupting tumor vascular supply and thus extirpates the spreading of tumor microthrombi, if they ever exist, within the supplying portal vein. However, HCC is a hypervascular tumor that can promote neoangiogenesis in the local tumor microenvironment, which in itself can break through the anatomical boundary within the liver and even retrieve nourishment from extrahepatic vessels, such as inferior phrenic or omental arteries. Additionally, increasing clinical evidence for locoregional tumor ablation therapies, such as radiofrequency ablation, predominantly performed as a non-anatomical approach, suggests comparable outcomes for surgical resection, particularly in small HCC and colorectal, hepatic metastases. Moreover, liver transplantation for HCC, which can be considered as AR of the whole liver followed by implantation of a new graft, is not universally free from post-transplant tumor recurrence. Overall, AR should not be considered the gold standard among all surgical resection methods. Surgical resection is fundamentally reliant on choosing the optimal margin width to achieve en-bloc tumor niche removal while balancing between oncological radicality and the preservation of postoperative liver function. The importance of this is to liberate surgical resilience in hepatocellular carcinoma. The overall success of HCC treatment is determined by the clearance of the theoretical niche. Developing biomolecular-guided navigation device/technologies may provide surgical guidance toward the total removal of microscopic tumor niche to achieve superior oncological outcomes.

## 1. Introduction

### 1.1. Surgical Goal of Hepatocellular Carcinoma

Hepatocellular carcinoma (HCC) is currently the third leading cause of cancer-related deaths and the fifth most common neoplasm in the world [1,2]. Curative interventions, including surgical resection, liver transplantation, and radiofrequency ablation (RFA), have been recommended for primary treatment [3]. Each of these approaches, if applied in adequately selected patients, could potentially offer a long-term survival benefit [4,5]. Treatment decision depends not only on tumor stages and anatomical locations but also on the patient’s sustenance of liver function [6,7,8]. In patients without significant cirrhosis or portal hypertension, the Barcelona Clinic Liver Cancer (BCLC) staging system suggests resection as the treatment of choice for single or limited numbered HCC [1]. The therapeutic goal is to achieve long-term cancer-free survival by resecting the entire malignant tissue (tumor, satellite nodules, and tumor-adjacent parenchyma) while preserving sufficient non-tumorous liver parenchyma to prevent postoperative liver failure [6,9].

### 1.2. AR: To Be or Not to Be, That Is the Question

Surgical methods can be broadly classified into anatomical resection (AR) and non-anatomical resection (NAR) [6,10]. First introduced by Makuuchi et al. in 1985, AR is defined as systematic removal of a hepatic segment confined by tumor-bearing portal tributaries [11,12,13], which could be marked by injecting a dye into the relevant portal veins [12,14]. Liver resection is considered AR if the following conditions are met: adequate identification of the resection area by exposing the vascular landmarks (hepatic veins) of the segment and ligation of the Glissonean pedicles at their origin [15]. AR usually involves two or more hepatic segments, whereas NAR involves tumor removal with a margin width of the uninvolved tissue [10]. NAR may benefit patients with HCC having cirrhosis or a less well-preserved liver function. Therefore, NAR can be considered a parenchyma-sparing alternative strategy.

Comparative studies between AR and NAR have indicated that AR provides superior survival benefits by reducing the risk of tumor recurrence and improving overall survival in patients with solitary HCC [16]. A systematic review by Moris et al. suggested that AR provides improved overall survival in patients without cirrhosis [17]. However, Kang and Ahn critically reviewed the results of well-designed comparative studies and suggested no significant difference in improving recurrence-free survival following AR [6]. A nationwide cohort study in Japan that compared AR and NAR reported no survival difference between the two methods in elderly patients with an HCC of less than 3 cm [18]. Until now, no prospective randomized controlled trial has confirmed the survival benefit of AR [6]. The superiority between AR and NAR remains controversial.

AR is one of the strategies for achieving a curative goal and to guide surgical resection. However, facilitating improved survival through curative resection is not limited to AR. Instead of anatomical or non-anatomical, the free (anatomical and microenvironmental) margin is the major concern. Both micro- and macroscopic free margin clearance determines the HCC recurrence rate and survival outcomes.

### 1.3. Hypothesis

We hypothesize that surgical eradication of HCC should not be AR-restricted. Additionally, surgical eradication of HCC is based on adequate clearance of the tumor together with the surrounding microenvironment niche. Herein, we performed a data-driven debate from an in-depth, focused review.

## 2. Evaluation of the Hypothesis

### 2.1. Circulating Tumor Cells Can Be Everywhere

The rationale for AR is theoretically effective for eradicating the intrahepatic metastases of HCC through the removal of tumor-bearing portal territories [12,19]. However, circulating tumor cells (CTCs) can be found in HCC-feeding vessels other than the portal system. Sun et al. [20] reported that the percentages of CTCs detected in blood sampled from a peripheral vein, peripheral artery, hepatic veins, infrahepatic inferior vena cava, and portal vein before HCC resection were 68.5%, 45.2%, 80.8%, 39.7%, and 58.9%, respectively (Figure 1). Moreover, CTCs and circulating tumor micro emboli burden detected in hepatic veins and peripheral circulation, but not portal vein, prognosticated postoperative lung metastasis, and intrahepatic recurrence, respectively. In Qi LN et al.’s study, AR may be more beneficial than NAR only in patients with low CTC count. The balance between operative risk and prognostic benefit is more important than the resection method in high CTC count patients [21]. Recently, Hidaka et al. reported about the pathological aspect of anatomical liver resection and concluded that AR for HCC with micro portal invasion (vp1) did not influence the recurrence-free survival or overall survival rates after hepatectomy [22]. This pathological evidence is consistent with the hypothesis.

### 2.2. Tumor Neoangiogenesis Does Not Follow the Anatomical Rule

Additionally, HCCs can derive new arterial blood supply from liver segment boundaries [23,24,25] and even from extrahepatic vessels, such as the inferior phrenic artery, omental arteries, or intercostal arteries [23,24,25] (Figure 2).

As HCC neoangiogenesis is not anatomically bound and CTCs can be found in multiple vascular routes other than portal veins, the rationale for performing absolute AR is not sufficiently strong. Surgical eradication of HCC should be flexible and not AR-restricted. The decision of reasonable resection margin clearance should consider the surrounding microenvironment niche [20].

Therefore, HCC neoangiogenesis is not restricted by normal anatomical boundaries.

### 2.3. Opposing Evidence 1: Local Treatment by Radiofrequency Ablation

The oncologic benefit is not exclusive to AR. RFA, a non-anatomical tumor ablation treatment performed irrespective of the hepatic blood supply anatomically [26], has proven to be an effective curative treatment alternative for HCC. According to the BCLC staging treatment guideline, thermal ablation, such as RFA, is the curative treatment of choice for patients with early-stage (BCLC 0-A) hepatic tumors [27]. RFA can provide comparable survival outcomes for liver resection with lower complication rates, such as bleeding, bile leakage, and post-treatment liver failure in early-stage HCC [7,28,29]. The statement that AR provides superior survival outcomes and less recurrence might be presumably attributed to larger liver resection with a greater tumor-free margin width [12,16,17].

### 2.4. Opposing Evidence 2: Liver Transplantation

Another opposing evidence to the use of universal AR is liver transplantation, which could be considered as AR of the whole liver and a new graft implant. However, post-transplant HCC recurrence can still occur at a rate of 13–27% [30], even under stringent selection criteria [30,31,32]. When the scenario was narrowed down to partial hepatectomy, the survival benefits of AR versus NAR were superior in all HCC patients (cirrhotic and non-cirrhotic) but similar in only cirrhotic patients [17,33,34]. The evidence for improved outcome measures outside of non-cirrhotic HCC patients is limited [34].

Therefore, the curative outcome and recurrence mechanism cannot be fully explained by AR alone, and resection margin clearance warrants more implication on recurrence outcome.

### 2.5. Surgical Perspective of the Microscopic Tumor Border

Recent advancements in the understanding of tumor biology and microenvironment enable reconsideration of the surgical planning strategy from a broader perspective. Cha et al. investigated the interaction between the tumor microenvironment and resection margin in different gross types of HCC and found that patients with expanding and vaguely nodular HCC may safely undergo surgical resection with a narrow resection margin, and patients with gross types, such as nodular with perinodular extension, multinodular confluent, and infiltrative types, should preferably undergo surgical resection with a wider (more than 2 cm) resection margin considering their tumor microenvironment conditions, namely expression of beta-catenin, matrix metalloproteinase 9, and E-cadherin [35]. The primary goal of surgical resection for primary HCC is to achieve adequate oncological radicality. The decision to choose a non-AR procedure should be based on key factors, such as pre-existing liver disease, tumor burden, recurrence risk, and whether the outcome will be affected by the extent of resection [10]. The post-resection organ failure concern is observed not only in hepatic resection surgery but also in lung resection surgery. Lesser lung parenchyma resection, such as segmentectomy or wedge resection, is indicated for patients who have a compromised pulmonary reserve to prevent post-surgery respiratory failure rather than standard lung lobectomy [36,37].

### 2.6. The Pro-Tumorigenic Niche Counts

The key decision in liver resection involves determining the “optimal” amount of non-tumor parenchyma to be removed. Considering the pro-tumorigenic niche heterogeneity in adjacent “non-tumor” parenchyma, which may contribute to future HCC recurrence, the definition of resection margin clearance could additionally be viewed as en-bloc removal of the niche including “HCC will definitely develop” in addition to the infiltration border of current HCC cells. These findings warrant the development of a new surgical planning and navigation strategy.

## 3. Consequences of the Hypothesis and Further Discussion

### 3.1. The Issue of Free Margin, Wider the Better?

Historically, nodular HCC is round in shape and grows expansively by compressing the noncancerous liver parenchyma; nodular HCC often possesses a fibrous capsule with clear-cut margins instead of infiltrating the non-cancerous parenchyma [38] (Figure 3). Therefore, the surgical resection margin (“tumor free” margin) surrounding the target HCC may not need that much. Shi et al. found that wide margin (2 cm) resection in HCC resection showed similar survival outcomes compared with narrow margin (1 cm) resection [39]. Additionally, Oguro et al. compared the patients who received macroscopic no-margin hepatectomy with those who received hepatectomy with macroscopic margin and found no difference in the recurrence-free and overall survival between the two groups [40]. By contrast, a moderate free margin for a poorly demarcated small tumor, which historically can be classified as a massive type [41], is necessary if the functional residual liver reserve is deemed acceptable. AR or non-AR, in these circumstances, may have a minor effect on HCC recurrence and patient survival.

### 3.2. Biological “Safe Margin”

Hoshida et al. [42] identified the poor prognosis gene signature driven by late recurrence originating from the adjacent cirrhotic tissue in patients with early HCCs, and the signature reflected the presence of a protumorigenic microenvironment (“field effect”) with metachronous tumor-promoting effects independent of the primary resected HCC. HCC develops from chronically damaged tissues that contain an extensive accumulation of inflammation and fibrosis, which promotes tumor progression. The poor prognosis signature from the adjacent tissue suggests that specific changes within the microenvironment affect the progression of HCC [43]. Therefore, the idea of safety margin in liver resection can be viewed as the en-bloc removal of the potential niche, wherein HCC is developing in the microscopic cellular scale.

### 3.3. Resection Planning Based on Tumor Niche Concept

An ideal “navigation” guiding tool should identify the protumorigenic niche while being cost-effective and convenient to use. Several attempts have been made to determine the “safety” margin of liver resection both clinically and experimentally. Andrea Peloso et al. [44] proposed the combined use of intraoperative sonography and indocyanine green (ICG)-fluorescence imaging, which can more efficiently detect small tumor lesions less than 3 mm in size than sonography alone. Ishizawa et al. [45] observed that pre-operatively injected ICG stasis remains longer in the HCC tumors than in normal hepatocytes, leading to tumor dying contrast enhancement during the surgery. This phenomenon is attributed to biliary excretion defects in cancerous tissues, causing the accumulation of ICG [46,47]. However, the tumorous fluorescence manifest specifically in well- or moderately-differentiated HCCs, whereas poorly-differentiated HCCs and metastases exhibit a rim fluorescence pattern. This inconsistency, the post-hoc nature of confirming tumor differentiation, and the thickness detection-depth limitation of 5–10 mm deep from the surface [45] prevent broad clinical application.

### 3.4. The Real-Time Navigator Probe: Under Developing

Another attempt in improving surgical guidance has been the development of an intraoperative detection probe similar to the sentinel lymph node detection device during breast cancer surgery. By combining specific markers binding to the HCC receptor, the protumorigenic niche could potentially be identified. In an in vivo xenograft mouse model, Wang et al. [48] used a cyclooxygenase-2-specific probe to identify the tumor region by emitting fluorescence binding to human HCC lines. Zhao et al. [49] used the novel glypican-3, a membrane-bound heparin sulfate proteoglycan that is highly expressed in HCC but negative in the normal liver tissue, to bind to the specific aptamer to target the glypican-3 positive HCC specifically (Figure 4). The aforementioned two studies claimed improvement in tumor-detecting accuracy in mouse models; however, neither of the studies presents sufficient evidence for clinical use. Other potential biomarkers for HCC border detection, such as expression of retinol [50] and DEPDC1 (DEP (Dishevelled/EGL-10/Pleckstrin) domain containing 1) [51], are emerging topics, and further investigations are warranted to validate the utility.

The limitation of this hypothesis-driven review is that the conclusions proposed in each cited paper are not all supported by hard data. Further studies are invited to confirm these conclusions by presenting the relevant data in the future.

## 4. Conclusions

In this article, we challenge the rationale for the universal adoption of AR for HCC. Curative resection is not limited to AR, and AR can present its own limitations. The concept of surgical resection should depend on choosing the optimal margin width for achieving en-bloc tumor niche removal, thus balancing oncological radicality and the preservation of postoperative liver function. Overall treatment success of HCC is determined by the clearance of the theoretical niche. Improved knowledge about cancer biology incorporating concepts of neoangiogenesis and micro-tumor biology, an emerging field of tumor microenvironment detection and protumorigenic niche, indicates that a new perspective of redefining resection margin clearance and further refining navigation devices/technologies could provide surgical guidance toward resection radicality for achieving superior oncological outcomes.

## Figures and Tables

**Figure 1 medicina-57-00131-f001:**
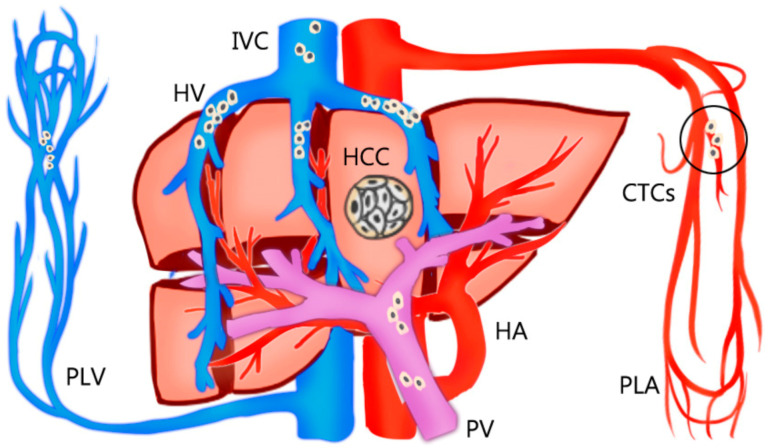
The percentages of circulating tumor cells (CTCs) detected in the bloodstream sampled from a peripheral vein (PLV), peripheral artery (PLA), hepatic veins (HV), inferior vena cava (IVC), and portal vein (PV) before resection of hepatocellular carcinoma (HCC). The circulating tumor cells are mostly detected in hepatic veins and peripheral circulation, but not portal vein.

**Figure 2 medicina-57-00131-f002:**
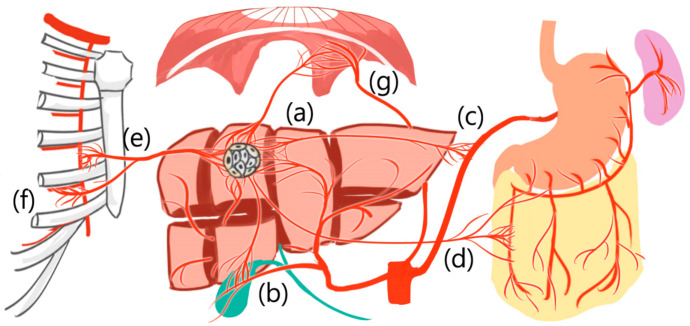
Hepatocellular carcinoma can derive new arterial blood supply from (**a**) adjacent liver segment and even from extrahepatic vessels, such as (**b**) cystic artery, (**c**) splenic trunk, (**d**) omental artery, (**e**) internal thoracic artery, (**f**) intercostal artery, and (**g**) inferior phrenic artery.

**Figure 3 medicina-57-00131-f003:**
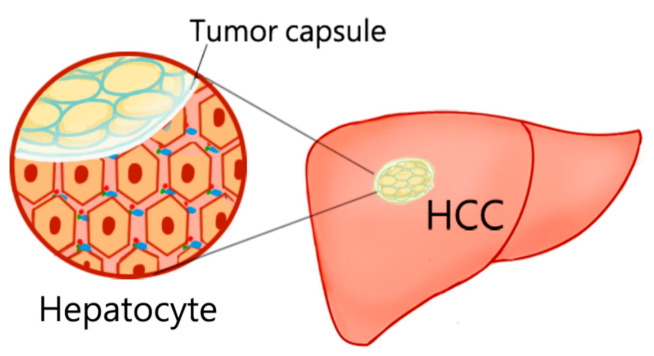
Hepatocellular carcinoma (HCC) often possesses a fibrous capsule with a clear-cut margin rather than an infiltrating border. The surgical resection margin (“tumor free” margin) surrounding the target HCC may not need that much in the former case.

**Figure 4 medicina-57-00131-f004:**
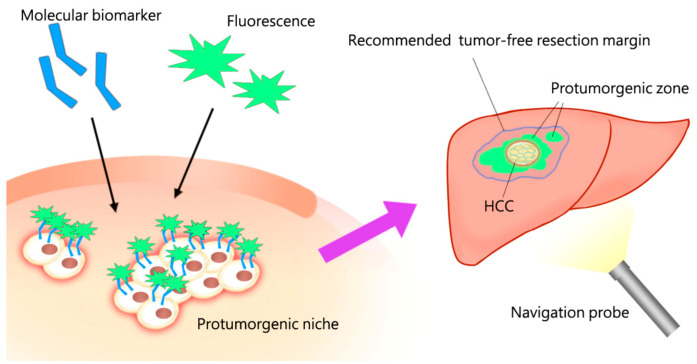
The redefined tumor free margin determined by specific biomolecular detectors, rather than by strict anatomical blood supply, could assist surgical planning to remove the protumorigenic niche en-bloc. This scenario, incorporating the principle concept of surgical oncology, is an emerging trend which can be derived from this hypothesis-driven review.

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
