# Peer review of "Is the Rationale of Anatomical Liver Resection for Hepatocellular Carcinoma Universally Adoptable? A Hypothesis-Driven Review"

_medicina, 2021, doi:10.3390/medicina57020131_

Round 1

Reviewer 1 Report

This is very important and comprehensive review about rationale for anatomical liver resection for hepatocellular carcinoma. It looks to be very informative and I do agree the whole hypothesis.

Recently, Hidaka et al reported about pathological aspect of anatomical liver resection and concluded that AR for HCC with vp1 did not influence the RFS or OS rates after hepatectomy (Ann Surg 2020;271:339–346). It will be much helpful for readers if you add this pathological evidence.

Author Response

Thank you very much for the positive comments and suggestion. We had added this important reference for pathological evidence.

Page 6.

Recently, Hidaka et al reported about pathological aspect of anatomical liver resection and concluded that AR for HCC with microportal invasion (vp1) did not influence the recurrence-free survival or overall survival rates after hepatectomy [22]. This pathological evidence is consistent with the hypothesis.

Ref 22

Hidaka M, Eguchi S, Okuda K, et al. Impact of anatomical resection for hepatocellular carcinoma with microportal invasion (vp1): A multi-institutional study by the Kyushu Study Group of Liver Surgery. Ann Surg 2020;271:339–346.

Reviewer 2 Report

The authors aim to scientifically evaluate the validity of anatomical resections ("AR") as a surgical treatment of hepatocellular carcinoma (HCC).
The topic is dealt with in a critical way, presenting the different alternatives that have been proposed over the years.
The bibliography that supports the topics of the paper is eminent and fairly up-to-date.
However, the conclusions proposed in each paper are not supported by hard data, based on the numbers reported by those papers. There are no statistical evaluations neither on the populations studied, nor on the results obtained. It would be interesting to confirm these conclusions by presenting the relevant data.
Furthermore, only one work cited was published in 2019, while all the others are prior to this date.
An evaluation of more updated data and experiences can enrich the paper.
Finally, the presentation of data collected by the authors could be useful to understand and increase the availability of data in the literature.

Author Response

However, the conclusions proposed in each paper are not supported by hard data, based on the numbers reported by those papers. There are no statistical evaluations neither on the populations studied, nor on the results obtained. It would be interesting to confirm these conclusions by presenting the relevant data.

Ans: Thank you for the precious comments. We agree that the conclusions proposed in each paper are not supported by hard data. We added a limitation statement before conclusion.

Page 11

The limitation of this hypothesis-driven review is that the conclusions proposed in each cited paper are not all supported by hard data. Further studies are invited to confirm these conclusions by presenting the relevant data in the future.

Furthermore, only one work cited was published in 2019, while all the others are prior to this date. An evaluation of more updated data and experiences can enrich the paper.

Ans: Thank you for the comments. We had updated some references status (Ref 21, 35, and 50). One important new reference 22 was added to enrich the paper.

Page 6.

Recently, Hidaka et al reported about pathological aspect of anatomical liver resection and concluded that AR for HCC with microportal invasion (vp1) did not influence the recurrence-free survival or overall survival rates after hepatectomy [22]. This pathological evidence is consistent with the hypothesis.

Page 16

  1. Hidaka M, Eguchi S, Okuda K, et al. Impact of anatomical resection for hepatocellular carcinoma with microportal invasion (vp1): A multi-institutional study by the Kyushu Study Group of Liver Surgery. Ann Surg. 2020;271:339–46.

Page 16

  1. Qi LN, Ma L, Chen YY, et al. Outcomes of anatomical versus non-anatomical resection for hepatocellular carcinoma according to circulating tumor-cell status. Ann Med 2020;52:21–31.

Page 18

  1. Cha SW, Kim SH, Kim YT, et al. Interaction between the tumor microenvironment and resection margin in different gross types of hepatocellular carcinoma. J Gastroenterol Hepatol 2020;35:648–53.

Page 19

  1. Han J, Han ML, Xing H, et al. Tissue and serum metabolomic phenotyping for diagnosis and prognosis of hepatocellular carcinoma. Int J Cancer 2020;146:1741–53.